# EMBED-SEARCH-ALIGN: DNA SEQUENCE ALIGNMENT USING TRANSFORMER MODELS

## ABSTRACT

DNA sequence alignment involves assigning short DNA reads to the most probable locations on an extensive reference genome. This process is crucial for various genomic analyses, including variant calling, transcriptomics, and epigenomics. Conventional methods, refined over decades, tackle this challenge in two steps: genome indexing followed by efficient search to locate likely positions for given reads. Building on the success of Large Language Models (LLM) in encoding text into embeddings, where the distance metric captures semantic similarity, recent efforts have explored whether the same Transformer architecture can produce numerical representations for DNA sequences. Such models have shown early promise in tasks involving classification of short DNA sequences, such as the detection of coding- vs non-coding regions, as well as the identification of enhancer and promoter sequences. Performance at sequence classification tasks does not, however, translate to *sequence alignment*, where it is necessary to conduct a genome-wide search to successfully align every read. We address this open problem by framing it as an "**E**mbed-**S**earch-**A**lign" task. In this framework, a novel encoder model *DNA-ESA* generates representations of reads and fragments of the reference, which are projected into a shared vector space where the read-fragment distance is used as a surrogate for alignment. In particular, DNA-ESA introduces: (1) Contrastive loss for self-supervised training of DNA sequence representations, facilitating rich sequence-level embeddings, and (2) a DNA vector store to enable search across fragments on a global scale. DNA-ESA is $> 97\%$ accurate when aligning 250-length reads onto a human reference genome of 3 gigabases (single-haploid), far exceeds the performance of 6 recent DNA-Transformer model baselines and shows task transfer across chromosomes and species.

## 1 BACKGROUND

The aim of this paper is to establish a foundation model tailored for DNA sequences, where the vocabulary consists of only a few symbols ($\{A, T, G, C\}$ in this case). Numerous DNA Transformer models (Ji et al., 2021; Zvyagin et al., 2022; Fishman et al., 2023; Dalla-Torre et al., 2023) have emerged recently, mainly designed for classification tasks in downstream applications. However, these models do not explicitly consider a fundamental distinction between Limited Vocabulary Languages (LVL) (such as, genomes) and natural languages with large vocabulary. In LVLs, there is a one-to-one correspondence between the precise symbol ordering and the underlying "meaning". If a protein is encoded by one amino acid sequence, a slightly different sequence (with a few edits) would encode for a different protein with a distinct functionality in the cells. Existing foundation Transformer models generate sequence embeddings such that their pairwise distances correspond to class separation, thus sequences with very large edit distances end up with representations that are close by. There are however tasks such as Sequence Alignment where the pairwise representation distance has to closely match the sequence edit distance. Indeed as portrayed in Figure 2, current DNA Transformer models fail to perform *Sequence Alignment* task. This requires a new kind of foundation model which is able to transfer a precise and computationally expensive distance metric over LVLs to a representation space.

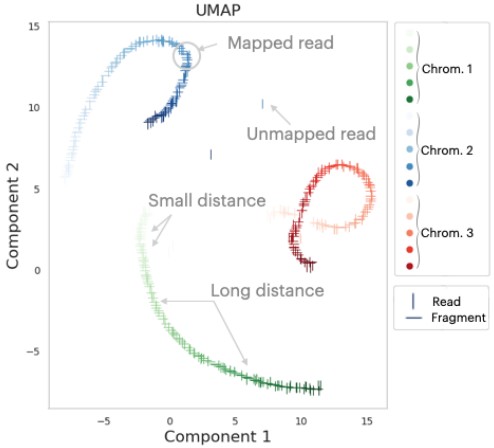 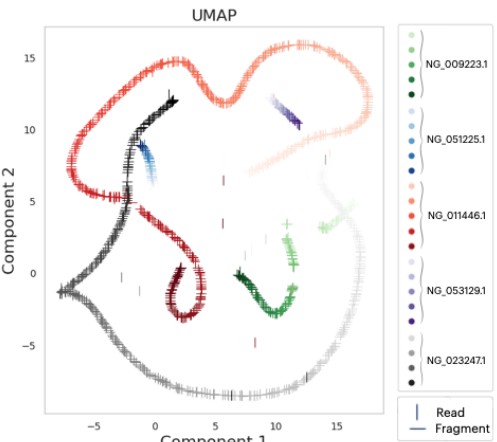

(a) A 20,000-long nucleotide sequence from Chr. 1, 2, and 3 each is randomly sampled and broken up into 100 *consecutive* reference fragments (|) each of length 1000 and stride 200. For each fragment, we sample a read (–) of 100-250 consecutive base pairs. Each sequence is encoded using DNA-ESA and the embeddings are visualized using 2D UMAP (McInnes et al., 2018). The colors identify chromosomes and the position of the fragment is coded by its intensity.

(b) Fragment (|) and read (–) sequences from five gene regions (listed above in the inset) are similarly projected into a shared latent space. Note that for UMAP projections (both subplots (a) and (b)), the consecutive fragments belonging to the same nucleotide sequence constitute an order-preserving 1D manifold. A successfully aligned read is observed to be co-located with its corresponding fragment in the embedding space while a read further away is observed to be misclassified.

Figure 1: **Illustrating DNA-ESA's Preservation of Sequence Locality in Embedding Space:** In DNA-ESA, the reference genome $\mathcal{R}$ is divided into fragments $\mathcal{F}_i$, each represented by an embedding $h(\mathcal{F}_i)$. For effective sequence alignment, specific structures are expected in the embedding space: (1) Overlapping fragments $\mathcal{F}_i$ and $\mathcal{F}_j$ should have proximate embeddings; (2) Consecutive fragments forming a long sequence should correspond to a distinct manifold in the embedding space. Subfigures (a) and (b) display this emergent geometry, as visualized using DNA-ESA.

The simplest sequence alignment task applies to single-end[1] reads. Given a reference sequence $\mathcal{R} := \{b_1, b_2, \ldots, b_N\}$ – for the single-haploid human genome (Nurk et al., 2022), $N \approx 3$ gigabases (gb) – the primary objective is to identify the most probable start- and end-positions within this reference for a short DNA read,

$$r := \{\tilde{b}_q, \tilde{b}_{q+1}, \ldots, \tilde{b}_{q+Q}\}, \; Q \ll N, \; 1 \leq q \leq N - Q \tag{1}$$

which may contain mutations due to base insertions, deletions, and substitutions. Computational simulators have been developed to generate synthetic reads that have properties of real reads. These simulators mimic the read quality and characteristics produced by actual sequencing machines, thus providing a scalable means for validating new alignment approaches (Huang et al., 2012).

Conventional sequence alignment methods, such as those used in BWA-MEM, have evolved significantly from the basic Smith-Waterman (SW) approach, which computes similarity between a read $r$ and reference $\mathcal{R}$, but is computationally intensive for large genomes. Key advancements include: (a) *Sharding* the reference into smaller fragments for individual searching; (b) *Progressive search* using phylogenetic trees and distance heuristics for logarithmic complexity scaling; (c) *Compression techniques* like the Burrows-Wheeler transform (Li & Durbin, 2009), reducing search length; and (d) *Multi-core/thread implementations and database instantiations* for faster computations and improved data recall (Li, 2018; Vasimuddin et al., 2019; Langmead et al., 2018).

In this work we explore an alternative paradigm for aligning a read to a genome, drawing parallels with advancements in Natural Language Processing (NLP). Traditional NLP relied on rule-based techniques to diagram sentences into grammatical components, using dependency parse trees to discern semantic relationships. However, the introduction the introduction of models like the Trans-

---

[1]A DNA fragment is ligated to an adapter and then sequenced from one end only.

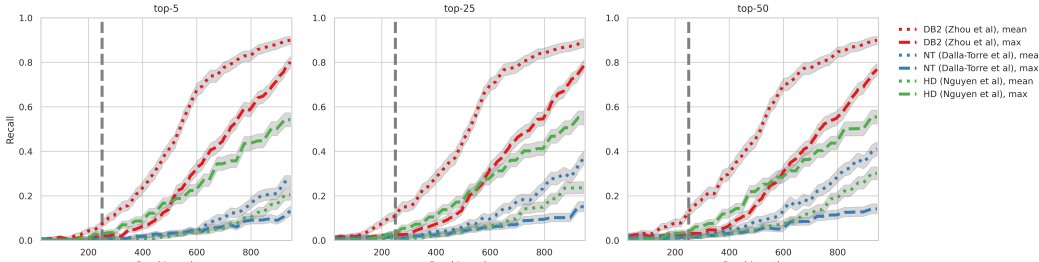

Figure 2: **Alignment Recall of Transformer-DNA Baselines by Read Length:** Existing Transformer-DNA models were adapted for sequence alignment using mean-/max-pooling. Their performance, measured by recall (top-$K$) over $40K$ reads of varying lengths across the human genome (3gb - single-haploid), is shown. Trendlines represent each baseline, with error bars (Clopper-Pearson Interval (Clopper & Pearson, 1934) @ 95%) in grey. The vertical line at $x = 250$ marks a typical read length. Overall, these baselines show suboptimal performance. For more details, see Sec. 4.

formers (Vaswani et al., 2017) revolutionized this approach. Unlike rule-based frameworks, Transformers implicitly identify syntactic and semantic structures, as evidenced in various NLP tasks like Sentiment Analysis, Entity Recognition, and Question-Answering (Devlin et al., 2019; Reimers & Gurevych, 2019; Qiu et al., 2020). The flexibility of this architecture has facilitated its application beyond language processing, including in vision (Dosovitskiy et al., 2020), auditory (Verma & Berger, 2021), and neurological domains (Wang et al., 2023). In bioinformatics, researchers are now harnessing Transformers to bypass genome-specific hard-coded rules, a development we contribute to with our novel method for Sequence Alignment.

## 1.1 Transformer models: Written Language to DNA Sequence Alignment

DNA sequences share remarkable similarities with written language, offering a compelling avenue for the application of Transformer models. Like written language, these are sequences generated by a small alphabet of nucleotides $\{A, T, G, C\}$. Classical DNA modeling efforts have already accommodated mature encoding and hashing techniques initially developed for written language – such as Suffix trees/arrays and Huffman coding (Huffman, 1952; Manber & Myers, 1993) – to successfully parse and compress DNA sequences. Furthermore, just as written language contains repeated subsequences (words, phrases) to represent real-world objects, DNA sequences similarly possess repeating "words" and groupings of such words into a "sentence" representing, for example, genes.

Within the last few years, several Transformer-based models have been developed for DNA sequence analysis. Notably, DNABERT-2 (Ji et al., 2021; Zhou et al., 2023), Nucleotide Transformer (Dalla-Torre et al., 2023), GenSLM (Zvyagin et al., 2022), and GENA-LM (Fishman et al., 2023) have been designed to discern relationships between short genetic fragments and their functions. Specifically, Nucleotide Transformer representations have shown utility in classifying key genomic features such as enhancer regions and promoter sequences. Similarly, GENA-LM has proven effective in identifying enhancers and Poly-adenylation sites in Drosophila. In parallel, DNABERT-2 representations have also been found to cluster in the representation space according to genetic function. Given these advances, a natural question arises: Can these Transformer architectures be readily applied to the task of Sequence Alignment? We delineate the associated challenges as follows:

**[L1] Two-Stage Training:** DNA-based Transformer models typically undergo pretraining via a *Next Token/Masked Token Prediction* framework, a method originally developed for natural language tasks. To form sequence-level representations, these models often employ pooling techniques that aggregate token-level features into a single feature vector. This approach, however, is sometimes critiqued for yielding suboptimal aggregate features (Reimers & Gurevych, 2019).

**[L2] Computation Cost:** The computational requirements for Transformer models grow quadratically with the length of the input sequence. This is particularly challenging for sequence alignment tasks that necessitate scanning entire genomic reference sequences.

Figure 2 shows the sequence alignment performance (recall) of several Transformer-DNA models. The testing protocols are elaborated in Sec. 4. Notably, these models exhibit subpar recall performance when aligning typical read lengths of 250.

## 2 OUR CONTRIBUTIONS

In this paper, we argue that both limitations **L1, L2** of Transformer-DNA models can be mitigated by formulating sequence alignment as a vector search-and-retrieval task. Our approach is twofold: (A) We introduce a sequence encoder *DNA-ESA*, trained through self-supervision, to map DNA reads to relevant fragments in a reference sequence within a shared embedding space. (B) We leverage a specialized data structure, termed a *DNA vector store*, as a memory bank. This provides efficient access to the entire reference sequence for each read alignment. These strategies have been explored in NLP: (A) Sequence-to-embedding training using contrastive loss has shown improved performance – over explicit pooling methods – at abstractive semantic tasks such as prose summarization and paragraph classification (Gao et al., 2021; Chen et al., 2020a). (B) Specialized data structures, such as "vector stores" or "vector databases" like FAISS (Johnson et al., 2019) and *Pinecone*, use advanced indexing and retrieval algorithms for scalable numerical representation search.

## 3 METHODS

We formulate the problem of Sequence Alignment as minimizing a sequence alignment function, SA, applied to a read $r$ and a reference sequence $\mathcal{R}$ as

$$v^* = \min_q \texttt{SA}(r, \mathcal{R}) \tag{2}$$

where $q \in \mathbb{N}_0$ is a candidate reference starting position and $v^*$ is the optimal alignment score. Lower scores indicate better alignments. This optimization exhibits the following property:

**[P1] Sharding for sequence alignment:** for a read segment $r$ of length $Q$ and reference $\mathcal{R}$ of length $N$, the complexity of $\texttt{SA}(r, \mathcal{R})$ scales as $\mathcal{O}(Q)$ when $N \to Q$.

Using **P1**, we can simplify the optimization problem by breaking it into sub-tasks with significantly shorter reference sequences. Specifically:

$$v^* \approx \min_{\mathcal{F}_i \in \{\mathcal{F}_1, \mathcal{F}_2, \ldots, \mathcal{F}_K\}} \texttt{SA}\left(r, \mathcal{F}_j\right). \tag{3}$$

Here, each $\mathcal{F}_j$ is a fragment of $\mathcal{R}$ (i.e., $\mathcal{F}_j \in \mathcal{R}$), and $K$ is the number of these sub-tasks[2]. This approximation is effective under the conditions:

(1) Fragment $\mathcal{F}_j$ lengths are on the order of the read length ($r$), not the longer reference ($\mathcal{R}$);
(2) There are enough fragments $\mathcal{F}_i$ to cover $\mathcal{R}$, i.e. $\cup \mathcal{F}_j = \mathcal{R}$;
(3) $K$ is significantly smaller than $\frac{N}{Q}$. If $\frac{N}{Q}$ then this amounts to scanning the whole reference.

Conditions (1) and (2) imply that fragments should be short and numerous enough to cover the reference genome. Condition (3) restricts the number of retrieved reference fragments per read — that we deem to be most likely to contain $r$ — to a small value $K$. Analogous methods have shown efficacy in text-based Search-and-Retrieval tasks (Peng et al., 2023; Dai et al., 2022) on Open-Domain Question-Answering, Ranking among other tasks. Subsequent sections describe a parallel framework for retrieving reference fragments given a read. The pipeline is shown in Figs. 3 and 4.

### 3.1 DESIGNING EFFECTIVE SEQUENCE REPRESENTATIONS

An optimal sequence encoder model $h$ is such that the corresponding embeddings of any read $r$ and reference fragment $\mathcal{F} - h(r), h(\mathcal{F})$ respectively – obey the following constraints over a predetermined distance metric $d$:

$$d\{h(r_j), h(\mathcal{F}_i)\} \geq d\{h(r_j), h(\mathcal{F}_j)\}, \quad i \neq j \tag{4}$$

---

[2]We denote the relative distance between an alignment and the optimal score at read length $Q$ as $d_{SW} = \frac{|mQ - v^*|}{|mQ|}$ where $m(= -2)$ is the match score while computing the SW distance.

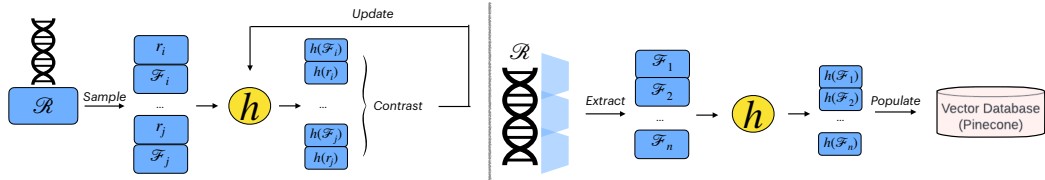

Figure 3: **System Overview [A] - Training Encoder and Populating Vector Store:** Reference genome fragments $\mathcal{F}_i$ and within them, randomly sampled pure reads $r_i$ (positive pairs) are numerically represented via shared encoder $h$. Encoder training follows a contrastive approach as per equation 5. After training, the genome is segmented into overlapping fragments, encoded, and uploaded into the vector store.

Here $i$ and $j$ serve to distinguish whether a read is aligned to a particular reference fragment, a *positive* sample $\{r_j, \mathcal{F}_j\}$, or there is a mismatch (*negative sample*): $\{r_j, \mathcal{F}_i\}$. Observe that these inequalities constitute the only requirements for the encoder. As long as the *neighborhood* of $r_j$ in the representation space *contains* the representation for $\mathcal{F}_j$, it will be recovered in the nearest neighbors (top-K set) and alignment will succeed. Equality is observed when $r_j$ is a repeat sequence matched equally well to more than one fragment. This motivates using self-supervision (Hadsell et al., 2006; Chen et al., 2020a; Gao et al., 2021) where we are only concerned about the relative distances between positive and negative (read, reference fragment) pairs.

## 3.2 SELF-SUPERVISION AND CONTRASTIVE LOSS

A popular choice for sequence learning using self-supervision involves a contrastive loss setup described by Chen et al. (2020a) and Gao et al. (2021): i.e. for a read $r$ aligned to reference fragment $\mathcal{F}_j$, the loss $l_r$ simultaneously minimizes the distance of $h(r)$ to $h(\mathcal{F}_j)$ and maximizes the distance to a batch of random fragments of size $B - 1$:

$$l_r = -\log \frac{e^{-d(h(r),h(\mathcal{F}_j))/\tau}}{e^{-d(h(r),h(\mathcal{F}_j))/\tau} + \sum_{i=1}^{B-1} e^{-d(h(r),h(\mathcal{F}_i))/\tau}}. \tag{5}$$

Here $\tau$ is a tuneable temperature parameter. To stabilize the training procedure and reach a non-trivial solution, the encoder applies different dropout masks to the reads and fragments similar to the method described in Chen et al. (2020b). Similar setups have been shown to work in written language applications, most notably in Sentence Transformers (Reimers & Gurevych, 2019; Gao et al., 2021; Muennighoff et al., 2023), which continue to be a strong benchmark for several downstream tasks requiring pre-trained sequence embeddings.

## 3.3 ENCODER IMPLEMENTATION

DNA-ESA uses a Transformer-encoder (Devlin et al., 2019; Vaswani et al., 2017), comprising 12 heads and 6-layers of encoder blocks. The size of vocabulary is $10,000$. Batch size $B$ is set to 16 with gradient accumulation across 16 steps. The learning rate is annealed using one-cycle cosine annealing (Smith & Topin, 2019), dropout is set to 0.1, and $\tau = 0.05$. Reference fragment $|\mathcal{F}_i| \sim \mathcal{U}[800, 2000]$ and read $|r_i| \sim \mathcal{U}[150, 500]$. Shorter sequences were padded to equal the length of the longest sequence in a batch. The distance metric used is *Cosine Similarity*.

## 3.4 SEARCH AND RETRIEVAL

An outline of the search and retrieval process is presented in Fig. 4. Every read is encoded using the trained model and matched to reference fragments in the vector database. The top-K retrieved fragments per read are then aligned using a SW alignment library to find the optimal alignment. The following sections describe the indexing and retrieval part in more detail.

**Indexing:** For a given reference genome $\mathcal{R}$, we construct a minimal set of reference fragments $\mathcal{F} := \mathcal{F}_1, \mathcal{F}_2, \ldots$ to span $\mathcal{R}$. Note that the fragments overlap at least a read length; i.e. $|\mathcal{F}_i \bigcap \mathcal{F}_{i+1}| \geq Q$ to guarantee that every read is fully contained within some fragment in the set. In our experiments with external read generators (Huang et al., 2012), $Q_{max} = 250, |\mathcal{F}_i| = 1250$. Each reference

Figure 4: **System Overview [B] - Inference on a New Read:** A read, as per equation 1 and generated by ART (Huang et al., 2012), is encoded by $h$. This is then compared to reference fragment representations in the vector database. The nearest-K fragments in the embedding space are retrieved for each read, and the optimal alignment is determined using equation 6.

fragment is encoded using the trained *DNA-ESA* model, and the resulting sequence embeddings ($\in \mathbb{R}^{384}$) – 3M vectors for a reference of 3B nucleotides – are inserted into a Pinecone database. Once populated with all the fragments, we are ready to perform the alignment.

**Retrieval:** Given a read $r$, we project its corresponding *DNA-ESA* representation into the vector store and retrieve the approximate nearest-$K$ set of reference fragment vectors and the corresponding fragment metadata $\{\mathcal{F}_1, \mathcal{F}_2, \ldots, \mathcal{F}_K\}$.

**Diversity priors:** While the top-$K$ retrieved fragments can be drawn from across the entire vector store (genome), contemporary recommendation systems that use the top-$K$ retrieval setup *rank and re-rank* top search results (*Slate Optimization* – see Zhu et al. (2007)) to ensure rich and diverse recommendations. Similarly, we apply a uniform prior wherein every retrieval step selects the top-$K$ *per* Chromosome.

**Fine-Alignment:** A standard SW distance library (Cock et al., 2009) is used to solve equation 3, which can be executed concurrently across the $K$-reference fragments. Let the optimal fragment be $\mathcal{F}^*$. The metadata for each vector includes (a) the raw $\mathcal{F}^*$ sequence; (b) the start position of $\mathcal{F}^*$ within the reference $\mathcal{R}$, $q_{\mathcal{F}^*|\mathcal{R}}$. Upon retrieval of a fragment and fine-alignment to find the fragment-level start index, $q_{|\mathcal{F}^*}$, the global reference start position is obtained as:

$$q^* = q_{|\mathcal{F}^*} + q_{\mathcal{F}^*|\mathcal{R}}.\tag{6}$$

## 4 TRANSFORMER-DNA BASELINES

This section outlines the setup for evaluating Transformer-DNA baselines, with their recall performance depicted in Fig. 2. We selected three architectures modeling nucleotide sequences: [NT] *NucleotideTransformer* ($\in \mathbb{R}^{1280}$) (Dalla-Torre et al., 2023), [DB2] *DNABERT-2* ($\in \mathbb{R}^{768}$) (Ji et al., 2021), and [HD] *HyenaDNA* ($\in \mathbb{R}^{256}$) (Nguyen et al., 2023). Each model employs mean- and max-pooling of token representations for sequence encoding ($2 \times 3 = 6$ baselines total). Independent vector stores for each baseline encode fragments from the entire 3gb genome. We sampled 40K pure reads of varying lengths ($Q \sim \mathcal{U}[25, 1000]$) and assessed the average recall for top-5, top-25, and top-50 fragments, as shown in Fig. 2. Overall, while baseline performance is modest, mean-pooling generally outperforms max-pooling, with DB2 (mean-pooled) and HD (max-pooled) as the most effective. These two baselines will be contrasted with DNA-ESA in Table 1.

## 5 RESULTS AND DISCUSSION

DNA-ESA convergence plots are presented in Figs. 5a and 5b. Model checkpoints are available at OSF. In Fig. 1, representations of short $1,000-$length sequences sampled from sequential (in-order) and gene-specific locations in the reference are visualized in a reduced 2D-UMAP (McInnes et al., 2018). The representation space demonstrates desired properties suitable for successfully performing alignment: (a) Sequences sampled in order form a trajectory in the representation space: The loss function described in equation 5 encourages a pair of sequences *close* to one another to have a short distance between them in the representation space, and pairs further apart to have a larger distance. (b) Representations of sequences drawn from specific gene locations – despite not being close to one another – show gene-centric clustering: The DNA-ESA representation space

| | | DNA-ESA (ours) | | | DB2, mean | HD, max |
|---|---|---|---|---|---|---|
| I | D | Top-$K$ @ 50 / Chr. | $d_{SW} = 1\%$ | Best | Top-K @ 50 / Chr. + $d_{SW} = 1\%$ | |
| | | | $Q_{PH} \in [60, 90]$ | | | |
| 0 | 0 | 95.2 ± 0.63 | + 1.6 | **96.8 ± 0.52** | 37.4 ± 1.35 | 14.6 ± 1.01 |
| 0 | 0.01 | 95.8 ± 0.59 | + 1.3 | **97.1 ± 0.50** | 37.4 ± 1.35 | 14.8 ± 1.01 |
| 0.01 | 0 | 95.1 ± 0.63 | + 2.1 | **97.2 ± 0.49** | 35.9 ± 1.35 | 14.6 ± 1.01 |
| 0.01 | 0.01 | 95.8 ± 0.59 | + 1.2 | **97.0 ± 0.51** | 34.9 ± 1.34 | 14.5 ± 1.01 |
| | | | $Q_{PH} \in [30, 60]$ | | | |
| 0 | 0 | 95.2 ± 0.63 | + 1.4 | **96.6 ± 0.54** | 35.4 ± 1.34 | 13.7 ± 0.98 |
| 0 | 0.01 | 95.1 ± 0.64 | + 2.0 | **97.1 ± 0.49** | 37.3 ± 1.35 | 14.2 ± 0.99 |
| 0.01 | 0 | 94.2 ± 0.68 | + 2.1 | **96.3 ± 0.56** | 36.0 ± 1.35 | 13.8 ± 0.98 |
| 0.01 | 0.01 | 94.9 ± 0.64 | + 1.5 | **96.4 ± 0.55** | 36.8 ± 1.35 | 13.5 ± 0.98 |

Table 1: **Performance of DNA-ESA with respect to baselines:** *(with diversity priors – see Def. 3.4)* The performance of DNA-ESA approeaches that of conventional algorithmic and far exceeds the performance of Transformer-DNA baselines, DB2, mean and HD, max (top performing baselines from Fig. 2). Both DNA-ESA and baselines utilize a dedicated vector store for the entire genome, maintaining a consistent search strategy. For details on $I, D, Q_{PH}, K, d_{SW}$, refer to *Recall/Simulator Configurations*.

partially acquires function-level separation as a byproduct of imposing *local* alignment constraints. Codebase is *linked*.

## 5.1 SEQUENCE ALIGNMENT OF ART-SIMULATED READS

The results from Sec. 4 demonstrate that even for pure reads, baseline models do not generate adequate representations to perform sequence alignment. In this section, DNA-ESA and the two best baselines – *DB2, mean* and *HD, max* – are evaluated using reads generated from an external read simulator (ART) – see Huang et al. (2012). ART has served as a reliable benchmark for evaluating other contemporary alignment tools and provides controls to model mutations and variations common in reads generated by Illumina machines.

**Simulator configurations:** The different simulation configuration options and settings are listed: (A) *Phred quality score $Q_{PH} \in \{[30, 60], [60, 90]\}$*: the likelihood of errors in base-calls of a generated read; (B) *Insertion rate $I \in \{0, 10^{-2}\}$*: the likelihood of adding a base to a random location in a read; (C) *Deletion rate $D \in \{0, 10^{-2}\}$*: the likelihood of deleting a base in the read; (Others): *Simulator system*: MSv3 [MiSeq]; *Read length*: 250.

**Recall configurations:** Once the top-K fragments have been retrieved, the first step is to solve equation 3, and for this, we need to compute the SW distance. For all presented results, the settings are: `match_score` = -2, `mismatch_penalty` = +1, `open_gap_penalty` = +0.5, `continue_gap_penalty` = +0.1. After alignment, we get $q^*$ – see equation 6 – as the estimated location of a read in the genome. Let $\hat{q}^*$ be its true location. If $q^* = \hat{q}^*$, it is a perfect match and the recall is successful. In cases where there is a mutation in the first or last position in a read, the fine-alignment will return $q^*_{\mathcal{F}^*}$ offset by at most 2 locations, resulting in $\hat{q}^* = q^* \pm 2$. Hence, the condition for an exact location match: $|q^* - \hat{q}^*| \leq 2$.

*Distance bound, $d_{SW} \in \{\text{None}, 1\%\}$*: It is well known that short fragments frequently repeat in the genome (Li & Freudenberg, 2014) and $q^*$ can correspond to the position of the read in a different location than from where it was sampled. In this case, $q^* \neq \hat{q}^*$, but the SW distance is the minimum possible ($d_{SW} = 0$ – see Footnote 2). Moreover, when reads have mutations, the reference sequence corresponding to the read is no longer a perfect match; i.e. $d_{SW} > 0$. The best an alignment algorithm could do is to find an exact match for the read leading to an optimal alignment score for that read length ($-2Q$). We consider an alignment (with $Q = 250$) to be successful if $d_{SW} < 0.01$; i.e. a mismatch of at most 2 bases.

**Performance:** Table 1 contains the result of sweeps along these several parameters in addition to a direct comparison to DB2, mean and HD, max baselines, the best-performing baselines on pure

| Train | Reads | Acc. ↑ | F1 ↑ | P ↑ | | R ↑ | |
|---|---|---|---|---|---|---|---|
| | | 2 (seen in training) | | | | 3 (populated) | Y (populated) |
| Chr. 2 | Pure | 98.0±0.43 | 98.5±0.39 | 99.0±0.28 | **97.5±0.46** | 98.2±0.41 | 99.3±0.27 |
| | ART | 97.9±0.43 | 97.8±0.43 | 97.8±0.43 | **97.9±0.43** | 97.6±0.44 | 98.0±0.43 |

Table 2: **DNA-ESA Performance on Chr. 2 and Task Transfer to Chr. 3, Y:** DNA-ESA, trained on Chr. 2, is evaluated for aligning positive samples from Chr. 2 and negative samples from Chr. 3. Anticipated failure to align Chr. 3 reads (false positives) is due to significant SW distance. Metrics reported include weighted accuracy (Acc.), precision (P), F1-score (F1), and recall (R), for comparison with Table 1. [Grey] For Chr. 3 and Y, separate vector stores are populated using DNA-ESA representations learned using Chr. 2 {read,fragment} pairs. These results support the generalizing tendency of DNA-ESA to align sequences beyond the training reference.

reads in Sec. 4. We observe the following: (A) DNA-ESA demonstrates strong recall of $\sim 97\%$ across a variety of read generation and recall configurations described in Sec. 5.1; (B) Reads with less noise – high $Q_{PH}$, low $I, D$ – are more often correctly aligned; (C) The $d_{SW}$ bound adds $\sim 1.5 - 2\%$ in recall (to cover 5% of misalignments) – a $\sim 20\%$ boost. This suggests that in the cases of misses, while finding the exact index match, the retrieved reference fragments are still high-quality retrievals that differ by at most 2 bases; (D) While the precise method of (a) generating reads (ART settings, gapped, long vs. short, paired-end vs. single-end, etc.), (b) accounting for the variation in the reference (with or without mitochondrial DNA, single- or double haploid, etc.), (c) detecting a successful alignment (duplicates, distance criteria) vary considerably, DNA-ESA approaches the performance reported by mature algorithmic methods of StrobeAlign, BWA-Mem2, and Minimap (Sahlin, 2022; Vasimuddin et al., 2019; Li, 2018).

## 5.2 TASK TRANSFER FROM CHROMOSOME 2

Are these results indicative of DNA-ESA's adaptability to new genomic sequences, rather than a strict adherence to its training data? This would suggest the model's learning to solve the sequence alignment task rather than memorizing the genome.

**Experiment setup:** DNA-ESA is trained on Chrosomome 2 – the longest chromosome – and recall is computed on unseen chromosomes from the human genome (3, Y) (*inter-chromosome*) and select chromosomes from chimpanzee (2A,2B) and rat (1,2) DNA (*inter-species*). Reads are either pure or ART-generated – as in Sec. 5.1, with the following simulator configurations: $I = 10^{-4}$, $D = 10^{-4}$, $Q_{PH} = [60, 90]$, $d_{SW} = 1\%$, $Q = 250$. Top-$K$ is set to 50, reads per setting = 5,000. Independent vector stores are constructed for each chromosome; representations for reference fragments (staging) and reads (testing) *are generated by the Chr. 2-trained model*. The results are reported in Tables 2, 3.

**Performance:** Details on convergence are in Appendix A. Table 2 shows that the performance on unseen human chromosomes (3, Y) is similar to the performance reported in Table 1, despite only training DNA-ESA on Chr. 2. This suggests DNA-ESA's ability to generalize sequence alignment across chromosomes with different compositions. High Accuracy, Precision, and F1-scores further confirm the method's specificity at this high recall. In Table 3, a trend of decreasing recall across species is observed: Human (2, 3, Y) > Chimpanzee (2A, 2B) (Blue) > Rat (1, 2), with chimpanzees likely performing better due to genetic similarities with humans (Ijdo et al., 1991). Even distantly related species like Thermus Aquaticus and Acidobacteriota show significant recall, highlighting DNA-ESA's task transferability beyond simple data memorization.

## 5.3 ABLATION STUDIES

We've conducted additional experiments on DNA-ESA, detailed in the Appendix: (A) Sec. C.1 covers DNA-ESA's performance on noisy reads from ART simulators and the PacBio CCS dataset. (B) In Sec. C.2, we explore its performance under various $d_{SW}$ bounds ({5%,10%}) and top-$K$ settings ({10,20}) per chromosome. High performance at lower $K$ values suggests a more effective representation space, while the $d_{SW}$ range assesses fragment retrieval accuracy without exact index matches. (C) Sec. C.3 describes outcomes when omitting diversity priors in searches across

| Species | Pure | ART |
|---|---|---|
| | Recall Top-K @50 ↑ | |
| *Thermus Aquaticus [All]* | 100.0 (93.0±0.75 @K=1) | 100.0 (95.2±0.63, @K=1) |
| *Acidobacteriota [All]* | 100.0 (93.2±0.74, @K=1) | 100.0 (94.9±0.64, @K=1) |
| *Rattus Norwegicus [Chr. 1]* | 93.3±0.73 | 95.4±0.62 |
| *Rattus Norwegicus [Chr. 2]* | 93.2±0.74 | 96.4±0.55 |
| *Pan Troglodytes [Chr. 2A]* | 94.5±0.67 | 97.1±0.51 |
| *Pan Troglodytes [Chr. 2B]* | 94.3±0.68 | 97.2±0.50 |

Table 3: **Cross-Species Task Transfer with DNA-ESA: Training on Human Chr. 2, Testing on Diverse Species:** DNA-ESA, trained on human Chr. 2, aligns fragments and reads from different species, including *Rattus Norwegicus, Pan Troglodytes, Acidabacteriota*, and *Thermus Aquaticus*. These species, are evaluated for read alignment recall using pure and ART-generated reads. The findings, consistent with Table 1, indicate DNA-ESA's proficiency in modeling DNA sequence structure, beyond simply memorizing training data. Top-$K = 1$ tests are conducted on species achieving 100% recall with $K = 50$.

the whole genome (top-$K$), resulting in predictably lower performance. (D) Sec. C.4 evaluates DNA-ESA's effectiveness with different read lengths, including those beyond $Q = 250$. Notably, performance improves with longer reads, demonstrating a *Zero-shot effect* for lengths not included in training.

# 6 FUTURE WORK

While DNA-ESA shows promising recall performance when compared to traditional algorithmic methods, its current computational speed, at 200 reads per minute, is a limiting factor for extensive genomic studies involving millions of reads. Despite this, the alignment task's inherent parallelizability (outlined in **[P1]**) offers avenues for efficiency improvements. Our ongoing and future efforts are directed towards overcoming this time efficiency bottleneck, as detailed in Appendix Sec. B. We are considering various optimization strategies, including model compilation to speed up inference and enhanced parallelization in vector store searches and fragment-read alignment. Additionally, we aim to enhance DNA-ESA's performance with shorter read lengths, exploring alternative training methods, integrating more diverse read/reference features, and employing data augmentation techniques.

# 7 CONCLUDING REMARKS

Current DNA sequence alignment methods rely on algorithmic approaches that have been refined over decades, incorporating DNA-specific enhancements in both indexing and retrieval. We have introduced an alternate data-driven paradigm. Employing a Transformer-based DNA-ESA encoder, our approach performs sequence alignment through self-supervised learning using contrastive loss. While such methods have previously been used for identifying (*approximate*) semantic similarity between sequences in written language applications, we demonstrate the surprising ability to find *exact* overlaps among DNA sequences. Our empirical results also show that the model, once trained, embodies the inherent structure of *any* DNA sequence (up to context length) irrespective of the location or species origin: a *foundation* model. DNA-ESA's innovation lies in its ability to represent various sequence lengths within a shared vector space, facilitating a 'flat' search approach. Unlike traditional methods that rely on hierarchical search, DNA-ESA treats shorter reads and longer reference fragments equally, using cosine similarity as a symmetric distance metric. This novel approach not only challenges existing paradigms but also enhances the potential for discovering novel DNA sequence representations. Mirroring the impact of Sentence Transformers in addressing sequence-level tasks in NLP, DNA-ESA, combined with traditional techniques, could lead to breakthroughs like aligning reads to the Pan Genome, where variations in the reference sequences model the genetic variations across individuals.

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

APPENDIX

## A    CONVERGENCE

Fig. 5 plot the convergence of the DNA-ESA encoder model discussed in the main text. We see convergence after $\sim$20k steps.

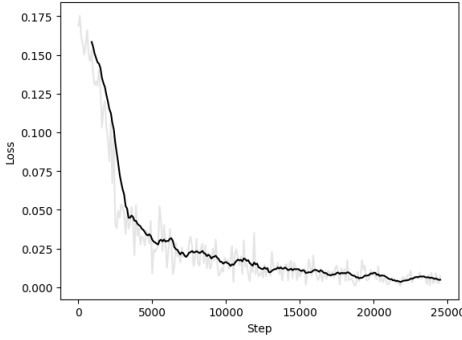
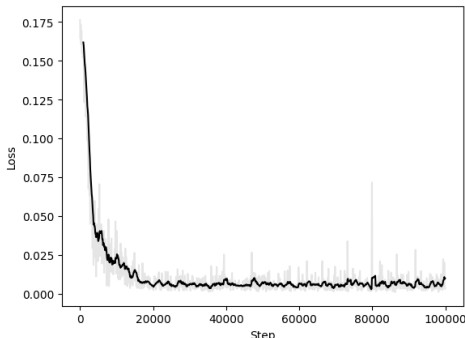

(a) Trained on the Human Genome  (b) Trained on the Human Chromosome 2

Figure 5: DNA-ESA convergence plots. Both plots show the loss pr. step (grey). For clarity, we smooth the loss using a moving average (black).

## B    COMPLEXITY OF COMPUTING ALIGNMENT

### B.1    COST OF CONSTRUCTING A NEW REPRESENTATION

Computing the embedding $\mathcal{E}$ of a sequence of length $F$ using DNA-ESA encoding – a typical Transformer-based attention architecture – has the following computation complexity:

$$\mathcal{O}(LH * (F^2 * d + d^2 * F)) \Rightarrow \mathcal{O}(F^2 * d + d^2 * F) \tag{7}$$

Where $d$ is the embedding dimension of the model, $L$ is the number of layers in the Transformer and $H$ is the number of heads per layer. As $d$ is a controllable parameter for the model, we can further simplify:

$$\mathcal{O}(F^2 * d + d^2 * F) \Rightarrow \mathcal{O}(F^2) \tag{8}$$

The $F^2$ complexity follows the basic implementation of attention in transformers, but recent efforts (Beltagy et al., 2020; Kitaev et al., 2020) have developed shortcuts to reduce the cost. These have already been applied to DNA sequence modeling (Nguyen et al., 2023).

### B.2    VECTOR STORE UPSTREAM

Vector store $\mathcal{D}$ is populated once (in bulk) with encoded fragment-length sequences drawn from the entire genome; *constant time complexity $C$ to upload $< 10M$ vectors.*

### B.3    RETRIEVAL COST

Given a new embedding, $\epsilon(G, K)$ is the cost of retrieving top-$K$ nearest neighbors across the fragment embeddings, where $G$ is the length of the reference genome. In modern vector databases, where several hashing techniques such as approximate K-nearest neighbors are used, $\epsilon$ scales logarithmically with $G$. This is indeed the key benefit of using such databases instead of a naïve search technique.

#### B.3.1    FINE-GRAINED ALIGNMENT

Existing libraries/algorithms (e.g. the Smith–Waterman algorithm) can identify the alignment between a fragment sequence (of length $F$) and read (of length $Q$) in $\mathcal{O}(FQ)$.

| Model | MiSeqv3 - Q = 250 (via ART, PH in [10,20], I,D:1%) | | | | PacBio, CCS - pbmm2, chr2 (Ashkenazim Trio, Son) | | |
|---|---|---|---|---|---|---|---|
| | Very-high Sensitivity | Alignment SW (500) | Very-high Specificity | Alignment SW (500) | Q=250 | Q=350 | Q=500 |
| *Bowtie-2* | **>99.9** | 490.2 | 87.0 ± 0.94 (–mp 1000 –np 1000) | 490.4 | <+2.0% | <+2.8% | <+2.5% |
| **DNA-ESA** | 94.4 ± 0.60 (dSW=30) | 488.3 | **89.0 ± 0.87** (@dSW=15) | **490.6** | 97.5 ± 0.47 | 96.5 ± 0.55 | 97.0 ± 0.51 |

Table 4: **DNA-ESA performance on noisy reads:** Noisy reads are simulated using ART ($Q_{PH} = 20$, $I, D = 1\%$) and derived from an external PacBio CCS read dump available in GIAB. The reads from PacBio are filtered to those derived from Chr. 2 as determined using the pbmm package Li (2018). In this evalaution, DNA-ESA is contextualized with Bowtie-2 Langmead et al. (2018), a popular conventional alignment tool. On both datasets, DNA-ESA performs slightly worse than Bowtie-2. Despite this, the quality of the retrieved fragments – as measured using the SW distance – is very similar. Note that in all cases, the performance of DNA-ESA is comparable (within 1%) to those reported in Table. 1.

### B.3.2 TOTAL COMPLEXITY

Total complexity involves (a) constructing the representation of a read; (b) querying the vector store; (c) running fine-grained alignment with respect to the $K$ returned reference fragment sequences:

$$\mathcal{O}(F^2 + FQK + \epsilon(G, K)) \Rightarrow \mathcal{O}(F(F + QK) + \epsilon(G, K)) \Rightarrow \mathcal{O}(FQK + \epsilon(G, K))$$

## C ABLATION STUDIES

### C.1 PERFORMANCE ON HIGH-NOISE READS

In Table 4, we report the performance of DNA-ESA on two noisy datasets: (i) ART-generated (MiSeqv3-based) reads of length $Q = 250$ with Phred quality score $Q_{PH} \in [10, 20], I, D = 1\%$ and (ii) reads from PacBio CCN as generated on the Ashkenazim Trio (Son) (retrieved from the Genome in a Bottle resource (GIAB) Zook et al. (2016)). For (ii), the raw reads are 10 kilobases long and for evaluation we consider random subset of 1000 reads derived to Chr. 2. Reads specific to a chromosome are filtered using the pbmm2 (Minimap) package Li (2018). Reads input into the aligner are cropped to length $250, 350$ and $500$. Note that in these experiments, we take care to increase the fragment-to-fragment overlap $|\mathcal{F}_i \bigcap \mathcal{F}_{i+1}|$ from 250 to 500. DNA-ESA performance is reported with respect to the Bowtie-2 performance. While Bowtie-2 has a $> 99.9\%$ recall, DNA-ESA isn't far behind and is comparable (within 1%) to the results on less noisy reads reported in Table 1. While the recall is lower, the quality of the recovered fragments is comparable: the average Smith-Waterman distance for all successfully aligned {read, fragment} pairs are similar.

### C.2 SWEEPING TOP-K AND SW DISTANCE BOUND

In Table 5, we report the performance of DNA-ESA with larger Smith-Waterman distance bounds $d_{SW} \in \{10\%, 20\%\}$ and a smaller number of recalled fragments $K \in \{10, 20\}$. Distance bound $d_{SW} < 10\%$ is equivalent to an acceptable mismatch of at most $\sim 8$ bases between the read (length $Q = 250$) and recalled reference fragments. We observe that with small $K$, the recall is $> 91\%$ (exact index match). As the distance bound increases, performance predictably improves; however, the performance gain from $d_{SW} = 25$ to $d_{SW} = 50$ is small. This implies that many retrieved fragments are high-quality retrievals; i.e. of high-likelihood to align with the read, and *do not benefit from* a more generous distance bound to improve recall performance significantly. Furthermore, decreasing the top-$K$ per chrosomosome from $20 \rightarrow 10$ does not substantially worsen performance ($\sim 2\%$) indicating that the optimal retrievals are usually the closest in the embedding space.

| $I$ | $D$ | | $Q_{PH} \in [30, 60]$ | | | $Q_{PH} \in [60, 90]$ | |
|---|---|---|---|---|---|---|---|
| | | Exact | $d_{SW} < 5\%$ | $d_{SW} < 10\%$ | Exact | $d_{SW} < 5\%$ | $d_{SW} < 10\%$ |
| | | | | Top-10 / chromosome | | | |
| 0.0 | 0.0 | 91.5±0.81 | 96.6±0.54 | 96.6±0.54 | 91.9±0.79 | 96.4±0.55 | 97.3±0.49 |
| | 0.01 | 92.0±0.79 | 96.5±0.55 | 96.8±0.52 | 92.0±0.79 | 96.2±0.57 | 97.6±0.47 |
| 0.01 | 0.0 | 91.9±0.79 | 96.4±0.55 | 96.5±0.55 | 92.0±0.79 | 96.6±0.54 | 97.4±0.48 |
| | 0.01 | 91.8±0.80 | 96.3±0.56 | 97.2±0.50 | 91.5±0.81 | 96.3±0.56 | 97.3±0.49 |
| | | | | Top-20 / chromosome | | | |
| 0.0 | 0.0 | 93.4±0.73 | 97.5±0.47 | 98.0±0.43 | 93.7±0.71 | 97.7±0.45 | 97.8±0.44 |
| | 0.01 | 93.2±0.74 | 97.2±0.49 | 98.2±0.41 | 93.6±0.72 | 97.5±0.47 | 97.8±0.45 |
| 0.01 | 0.0 | 93.0±0.74 | 97.9±0.44 | 97.9±0.43 | 92.6±0.76 | 97.4±0.48 | 97.8±0.44 |
| | 0.01 | 93.0±0.74 | 97.4±0.48 | 97.8±0.45 | 93.4±0.72 | 97.8±0.45 | 98.1±0.42 |

Table 5: **Sequence alignment recall of DNA-ESA sweeping top-K and $d_{SW}$:** The various parameters are described in Sec. 5.1. DNA-ESA presents a recall of $> 97\%$ across several read configurations rivaling contemporary algorithmic models. Performance predictably degrades as more noise is introduced into a read. Performance improves with larger search radius (top-$K$), higher quality reads ($Q_{PH}$) and large distance bound $d_{SW}$.

## C.3 WITHOUT DIVERSITY PRIORS IN TOP-K

In Table 6, we report the performance without diversity priors used in the retrieval step: i.e. the nearest-K neighbors in the embedding space are selected from the *entire* set of fragments spanning the genome rather than uniformly sampling from each chromosome. The performance predicably falls in comparison to those reported in Table 1 since fewer fragments scattered unevenly across the different chromosomes are being retrieved per read.

| $I$ | $D$ | | $Q_{PH} \in [30, 60]$ | | | $Q_{PH} \in [60, 90]$ | |
|---|---|---|---|---|---|---|---|
| | | Exact | $d_{SW} < 5\%$ | $d_{SW} < 10\%$ | Exact | $d_{SW} < 5\%$ | $d_{SW} < 10\%$ |
| | | | | Top-100 | | | |
| 0.0 | 0.0 | 82.6±1.08 | 92.0±0.79 | 93.2±0.74 | 88.5±0.92 | 91.9±0.79 | 93.0±0.74 |
| | 0.01 | 82.3±1.09 | 91.8±0.80 | 92.7±0.76 | 89.3±0.89 | 93.2±0.74 | 93.9±0.70 |
| 0.01 | 0.0 | 83.3±1.06 | 92.3±0.77 | 93.3±0.73 | 88.7±0.91 | 92.4±0.77 | 93.3±0.73 |
| | 0.01 | 82.8±1.08 | 91.6±0.81 | 92.5±0.76 | 88.7±0.91 | 92.4±0.77 | 93.2±0.73 |
| | | | | Top-50 | | | |
| 0.0 | 0.0 | 81.1±1.11 | 90.6±0.84 | 92.0±0.79 | 86.7±0.97 | 90.7±0.84 | 91.9±0.79 |
| | 0.01 | 80.5±1.13 | 90.4±0.85 | 91.6±0.80 | 87.4±0.95 | 91.7±0.80 | 92.7±0.76 |
| 0.01 | 0.0 | 81.3±1.11 | 90.8±0.83 | 92.3±0.78 | 86.8±0.97 | 91.0±0.83 | 92.2±0.78 |
| | 0.01 | 81.2±1.11 | 90.4±0.85 | 91.7±0.80 | 87.1±0.96 | 91.2±0.82 | 92.4±0.77 |

Table 6: **Sequence alignment recall of DNA-ESA - without diversity priors:** The various parameters are described in Sec. 5.1. DNA-ESA presents a recall of $\approx 90\%$. Similar to the result presented in the main text, performance improves with larger search radius (top-$K$), higher quality reads ($Q_{PH}$) and large distance bound ($d_{SW}$).

## C.4 ACROSS READ LENGTHS

In Table 7, the performance of DNA-ESA is reported across read lengths. We observe *Zero-shot performance at longer read lengths*: The model performs better at longer read lengths (even exceeding the read length bound established during training $\mathcal{U}[150, 500]$); while evaluating for longer reads, we make sure to guarantee that the reads exist as subsequences of fragments. Improving the performance at shorter read lengths is the subject of future work.

| | | | Top-K @ 50 | | |
|---|---|---|---|---|---|
| I | D | $Q = 150$ | $Q = 200$ | $Q = 250$ | $500 < Q < 1000$ |
| | | $Q_{PH} \in [60, 90]$ | | | Pure Reads |
| 0 | 0 | 84.6 ± 1.03 | 94.9 ± 0.65 | **96.8 ± 0.52** | >97 |
| 0 | 0.01 | 85.9 ± 0.99 | 95.2 ± 0.63 | **97.1 ± 0.50** | |
| 0.01 | 0 | 85.8 ± 0.99 | 94.5 ± 0.66 | **97.2 ± 0.49** | |
| 0.01 | 0.01 | 86.2 ± 0.98 | 95.3 ± 0.63 | **97.0 ± 0.51** | |
| | | $Q_{PH} \in [30, 60]$ | | | |
| 0 | 0 | 84.6 ± 1.03 | 94.6 ± 0.66 | **96.6 ± 0.54** | >96 |
| 0 | 0.01 | 85.2 ± 1.01 | 95.0 ± 0.64 | **97.1 ± 0.49** | |
| 0.01 | 0 | 85.7 ± 1.00 | 94.5 ± 0.66 | **96.3 ± 0.56** | |
| 0.01 | 0.01 | 85.8 ± 0.99 | 94.4 ± 0.67 | **96.4 ± 0.55** | |

Table 7: **DNA-ESA recall performance across read lengths:** Performance of DNA-ESA is higher for longer reads including those lengths on which the model was not trained ($Q \in \mathcal{U}[150, 500]$). Shorter reads are more challenging for the model potentially due to replicates found across the reference.

