# OpenReview forum: "Embed-Search-Align: DNA Sequence Alignment using Transformer models"
_ICLR.cc/2024/Conference — Submitted to ICLR 2024_

### Official Review · Reviewer_nXVu · 2023-10-26

**Soundness:** 1 poor
**Presentation:** 3 good
**Contribution:** 3 good
**Rating:** 3
**Confidence:** 4

**Summary:**

This paper introduces an embedding for DNA sequences based on transformer models. This embedding allows a distance between two DNS fragments of potentially differing lengths to be computed. Using this distance, an alignment algorithm is constructed under the usual seed-align framework, with the embedding distance taking the place of the seeding step. The embedding quality is assessed against other transformer models on synthetic reads.

**Strengths:**

The problem being tacked is an important one, and the proposed architecture does appear to have some benefits over existing transformer models. The ability to handle varying fragment lengths makes the method very flexible.

**Weaknesses:**

The experimental validation is the weak point of this paper. There are claims of efficiency yet no experimental evidence of any resource requirements and scaling such as time and memory. Furthermore, comparitive experiments are against existing transformer models and hence show that the embedding is superior to existing ones for alignment, but not that this is a good aligner overall. There are no comparisons against standard alignment algorithms (though they are referenced). There are no experiments on real data, or evaluation of the effect on downstream variant calling.

Generalisation to other species is very interesting, however the chosen species are all closely related.

The constraint in equation 4 cannot strictly hold on real data due to homology and repeat regions.

**Questions:**

- Have you evaluated the method on real data?
- What are the runtime resource requirements?
- Does the model generalise to less related organisms? Presumably there will be some dependence on degree of homology.

---

> ### Author Response · Authors · 2023-11-22
>
> We thank the reviewer for their comments and kindly point them to the general comments in addition to specific comments below. Kindly also review the responses to other reviewers.
>
> **[Q1] Have you evaluated the method on real data/data with noise?**
>
> Before we discuss the additional evaluation results we have included in the revised version, we wanted to address why we used ART as a simulation toolkit. An important consideration was that for such reads, we know the ground truth (making evaluation much more precise) while ensuring that the reads have the same distribution of errors as one would expect from real world sequencers.
>
> In fact, ART is an industry-standard simulator that is used to benchmark almost every new aligner that is introduced. ART mimics the variants, insertions/deletions (INDELs) and other error models found in reads generated by expensive Illumina machines. The ART simulator used in our experiments was fine-tuned on an Illumina MiSeqv3 machine model (https://www.illumina.com/systems/sequencing-platforms/miseq/specifications.html).
> BWA-Mem and several other aligners have been validated on such simulated data: Table 1 here: https://www.ncbi.nlm.nih.gov/pmc/articles/PMC2705234/ demonstrates performance on WGSIM from SAMTools, which is a more primitive version of ART (https://bmcbioinformatics.biomedcentral.com/articles/10.1186/1471-2105-14-184). ART was also used in the simulation study of the 1000 Genomes datasets to profile the quality of the resource: https://www.niehs.nih.gov/research/resources/software/biostatistics/art/index.cfm
>
> We also want to note that Pure Reads serve an important purpose: They establish an upper bound on the performance of the aligner (as noisy reads would only have lower performance), allowing us to benchmark the maximum possible performance for Transformer-based LLM models as shown in Fig. 2. These pure-read results make it superfluous to benchmark the existing Transformer models on noisy data / real data.
>
> In the revision: While we believe that ART reads with $Q_{PH} >= 30$  are sufficient to benchmark aligners -- Next-gen Illumina machines generate reads at Phred score $Q_{PH} >= 30$, we address the reviewers’ request by evaluating DNA-ESA on reads with $Q_{PH} \in [10,20]$, $I,D = 1$%; see Table 4 (left).
>
> We are pleased to report that DNA-ESA performance at $Q_{PH} \in [10,20]$, $I,D = 1$% is only about $1$% less than the performance reported at $Q_{PH} \in [30,60]$, $I,D = 0.01$%.
>
> Performance on real read data: In Table 4, we have also included the performance of DNA-ESA on real reads generated by PacBio CCS ($Q_{PH} = 20$): and read length $\in \{250, 350, 500\}$.
>
> Here, too, DNA-ESA performs as well on real read data as on ART-generated reads reported in our original submission.  This further supports our original evaluation platform.
>
>
> **[Q2] What are the runtime resource requirements?**
>
> The model requires a GPU with >10GB of vRAM to run inference on DNA-ESA. The model uses 20GB of CPU memory during inference. This is an initial version of the model, and as noted in Sec. Future Work, will continually become more performant using methods such as model distillation, compilation, and distributed processing techniques. Efficient transformers are still an area of active research which have seen notable improvements over the last few years (e.g. FastAttention, Long-range architectures such as HyenaDNA and similar).
>
> **[Q3] Does the model generalize to less related organisms? Presumably there will be some dependence on the degree of homology.**
>
> We thank Reviewer 3 for their curiosity in understanding whether the model generalizes to less related organisms. Accordingly, we have expanded the inter-species experiments in the transfer learning section (Table 3) to include Thermus Aquaticus -- an ancient extremophile bacterium -- and Acidobacteriota reference sequences: two species that are highly dissimilar to the rest. We are pleased to report that the model -- originally trained to align reads to fragments on Chr. 2 of humans -- continues to perform well (in fact even with top-K = 1, the recall is >93%)! This adds further support to the claim that the DNA-ESA architecture truly learns the alignment task rather than the sequences themselves seen during training.

---

### Official Review · Reviewer_o7rD · 2023-10-31

**Soundness:** 2 fair
**Presentation:** 2 fair
**Contribution:** 2 fair
**Rating:** 3
**Confidence:** 4

**Summary:**

This manuscript tries to solve an alignment problem. The input are a set of short DNA sequences (reads) and a long reference sequence. The task is to find the location of each short read on the reference sequence. This manuscript proposes to solve this matching problem by projecting both short reads and reference (sub-)sequence to a shared embedding space such that reads originated from the same reference sequence region are close to each other in the embedding space. For a new read, its embedding is first calculated using the NN, after that it searches for the nearest K reference fragments in the embedding space. Exact alignment using classical Smith-waterman algorithm is between the given read and each of the K nearest reference fragments to identify the final mapping position of the read.

Regarding implementation, the authors first split the reference sequence (length = 3Gb) into fragments (length = 1250bp). Then each fragment and their substrings (reads) are assigned the same label, which (a batch of several fragments and their substrings) is then fed into a transformer with a contrastive loss. The trained network can then project a given DNA sequence to the embedding space (dimension = 384).

Although I think the idea is somewhat interesting, the authors may want to address the following points in the next version.
1. Provide support to the hypothesis that a fragment and its subsequences are close in the embedding space. Figure 1 only contain the fragments, but it should also contain random subsequences.
2. Make the notations more readable, e.g. in Eq. 2, q is used in SA(r,R). Eq. 6, really difficult to guess the meanings.
3. Comparison with classic aligner such as BWA or bowtie. I will expect a >99.9% mapping rate of BWA or bowtie for the used data.
4. Expand the comparison dimensions. Only Recall is used, could add precision, mapping rate etc.
5. Compare on real data. The simulated data is too clean. Phred 30 is already 99.9% identical to the original sequence, that is 250/1000=0.25 bp mutation per read. It will be nice to see how the method works on real data where the error rate is higher and the read length varies.
6. Provide details of the transformer. Do you do paddings to short reads?
7. What happens to negative samples, i.e. a random read that is not similar to any reference fragment? How do you decide the boundaries of a reference fragment in the embedding space?
8. Reorder the figures so it flows with the text.
9. Appendix A, add legend for the lines
10. Appendix B, B.3 the complexity is G, not log(G) as you have to compare with all reference fragments.

**Strengths:**

NA

**Weaknesses:**

NA

**Questions:**

NA

---

> ### Author Response · Authors · 2023-11-22
>
> We urge the reviewer to kindly read the general comments as well as our replies to other reviewers.
>
> **[Q1] Provide support to the hypothesis that a fragment and its subsequences are close in the embedding space.**
>
> We agree with the reviewer and we have updated Figure 1 to visualize both reads and fragments. We see convincing visual evidence that both the reads and the fragments align within the embedding space.
>
> **[Q2] Make the notations more readable, e.g. in Eq. 2, q is used in SA(r,R). Eq. 6, really difficult to guess the meanings.**
>
> We have resolved this discrepancy and now use $q$ in Eq. 2 as well to denote the start position. Furthermore, for Eq. 6, we realize that the asterisks on $F$ and $q$ are difficult to read. So we have added clarity to the equation in the new version.
>
> **[Q3] Comparison to existing classical methods (Bowtie-2, Minimap, BWA-Mem-2)**
>
> We have presented the side-by-side results of Bowtie-2 and DNA-ESA in Table 4 on two noisy datasets. We are happy to report that the performance of Bowtie-2 is only 3-4% above the recall performance of DNA-ESA. The quality of the recalled fragments (of those reads that are successfully aligned) is comparable among the two models.
>
> **[Q4] “Provide details of the transformer. Do you do paddings to short reads?”**
>
> Clarified section 3.3: “Reads and shorter fragments were padded to equal the length of the longest sequence sampled in a batch.”
>
> **[Q5] What happens to negative samples, i.e. a random read that is not similar to any reference fragment?**
>
> We thank the reviewer for the insight about including additional metrics such as Accuracy, Precision and F1 score in addition to Recall: the classic case of evaluating specificity vs. sensitivity. We are happy to report in Table 2 that these metrics are all high: negative samples were drawn from Chr. 3 with positive samples from Chr. 2.
>
> **[Q6] How do you decide the boundaries of a reference fragment in the embedding space?**
>
> We are unclear about what the reviewer means by boundary. Our guess is that the reviewer asks for clarification about how we go to the exact location of a read from the top-K fragments. Note that each fragment has meta-data corresponding to the reference location from which the fragment is drawn.
> Below we also briefly recap the alignment process.
> (Step 1) Identify the long fragments corresponding to the most likely match to a read. Top-K neighbor is computed in log complexity; (Step 2) Fine-alignment of the read to the fragments: Compute Smith-Waterman distance between the read and each of the candidate fragments. This distance metric is thresholded. We sweep across several of these thresholds in Table 3 and describe the process in Appendix C.1.
>
> **[Q7] Reorder the figures so it flows with the text.**
>
> We extend our gratitude to the reviewer for their valuable comments. We hope to convince the reviewer that the current order is justified.
>
> Figure 1 plays a pivotal role by immediately illustrating the generated embedding space, which forms the foundation for all subsequent downstream applications. This depiction holds particular appeal for Machine Learning enthusiasts, as it showcases the training method's unique ability to generate trajectories within the embedding space.
>
> Figure 2 provides a side-by-side performance comparison of various Transformer-based models on pure reads (reads without any noise). Despite the apparent simplicity of this task, it becomes evident that competing Transformer-based models struggle to address it. Finally, with the motivation firmly established, Figures 3 and 4 delve into the intricacies of the method, describing the finer points of how these representations are generated.
>
> **[Q8] Appendix A, add legend for the lines**
>
> We have added the following to the figure text: “DNA-ESA convergence plots. Each plots show the loss pr. step (grey). For clarity, we smooth the loss using a moving average (black).”
>
> **[Q9] Appendix B, B.3 the complexity is G, not log(G) as you have to compare with all reference fragments.**
>
> *It's crucial to highlight that the search complexity across G is, indeed, logarithmic (log(G)).*
>
> This logarithmic scaling represents a fundamental advantage derived from utilizing vector stores in our task. Without the efficiency gained from vector comparisons, resorting to direct string representations for matching the read with every fragment might as well have worked.
>
> Vector stores opt for a highly accurate **approximate** K-nearest neighbor method in addition to other proprietary optimizations to search that include Small-World approximations, hierarchical organization of graph data and concurrent search. This is the entire reason behind FAISS’ ubiquity and Pinecone’s rapid success as a vector store solution. This innovation facilitates the breakthrough in our representation learning solution to alignment.

---

### Official Review · Reviewer_ModJ · 2023-11-03

**Soundness:** 3 good
**Presentation:** 3 good
**Contribution:** 3 good
**Rating:** 6
**Confidence:** 3

**Summary:**

The paper presents EMBED-SEARCH-ALIGN, a method for DNA sequence alignment using transformer models.  Unlike traditional methods use algorithmic solutions like indexing and efficient searching to align reads. This paper proposes a different paradigm using transformer models to learn representations of DNA reads and reference genome fragments, and performing alignment based on similarity of these embeddings.

The main contributions of the proposed architecture are as  follows. First, the authors use a DNA sequence encoder DNA-ESA that is trained using self-supervision and contrastive loss to produce DNA sequence embeddings optimized for alignment. Next, the authors use a DNA vector store to enable efficient nearest neighbor search across the entire reference genome for each read. Finally, the authors formulate the sequence alignment as an "embed-search-align" task using the encoder and vector store.

In the experiments, the authors demonstrate that DNA-ESA can align 250 bp reads to the human reference genome with over 97% accuracy, exceeding several transformer baseline models. The approach also demonstrates ability to generalize - a model trained only on Chr 2 can still align reads from other chromosomes and even other species.

The paper argues this approach mitigates limitations of prior works on transformer models for genomics and provides sequence representations suitable for alignment. It also enables "flat search" over reads and reference fragments of different lengths. Future work is discussed to improve computational efficiency.

**Strengths:**

+ The authors present a novel approach to align sequence reads which can provide new possibilities for DNA sequence representation and search.
+ The proposed DNA-ESA encoder learns effective sequence embeddings for alignment, and outperforms several baseline transformer models designed for specific genomics tasks.
+ The approach is promising and demonstrates ability to generalize to new sequences not seen during training, like different chromosomes and even new species. Furthermore, formulating alignment as embed-search-align could enable new capabilities like "flat search" over reads and reference fragments of different lengths.

**Weaknesses:**

- I felt that the paper is a very dense read for the general ML audience at ICLR for folks who do not have DNA sequencing background, and it will be great to make the paper more accessible.
- The embedding approach currently shows promising results on simulated data, but needs more evaluation on real sequencing data.
- The performance for short reads is worse than long reads, given that short reads are more commonly used, this may affect how this system can be actually used.
- Limited demonstration of applications in downstream genomic tasks.

**Questions:**

1) Can the authors comment on how this paper is a good fit for ICLR, and the steps the author may take to make this paper more accessible to the ML audience?
2) How does the model perform on real world sequencing data?

---

> ### Author Response · Authors · 2023-11-22
>
> We thank Reviewer 1 for their comments about our submission. We appreciate their recognizing the promise of the approach, specifically (a) the effectiveness of the sequence embeddings generated by DNA-ESA (as demonstrated in search); (b) the generalization across species and the flexibility of the approach in encoding reads and fragments of varying lengths.
>
> We urge the reviewer to also review the general comments in addition to our responses to other reviewers. Below we address the specific concerns:
>
> **[Q1] “I felt that the paper is a very dense read for the general ML audience at ICLR for folks who do not have DNA sequencing background, and it will be great to make the paper more accessible.”**
>
> Acknowledging the interdisciplinary nature of our work, we concur with the reviewer's perspective that it is our responsibility, as the authors, to bridge the gap between cutting-edge techniques in NLP and the longstanding challenges in bioinformatics and genomics. To address this, we have taken the following steps:
>
> In Eq. 1, we define a read in its simplest form: a sequence of bases. In Eq.s 2 and 3, we write the objective of sequence alignment through the application of the sharding rule [P1]. Furthermore, in Section 3.1, we present the rationale behind a straightforward inequality constraint Eq. 4 (modified now to incorporate homologs / repeats) within the representation space.
>
> It is worth noting that our approach to motivating sequence alignment, as outlined, is itself innovative and designed to resonate with the machine learning community. We remain optimistic that the reviewer finds merit in this novel perspective.
>
>
> **[Q2] “The performance for short reads is worse than long reads, given that short reads are more commonly used, this may affect how this system can be actually used.”**
>
> The reviewer rightly notes that short-read aligners are more commonly used than long-read aligners. Our approach does not intend to replace existing short-range alignment methods. Rather, methods for sequencing have seen rapid advances in cost and their ability to produce increasingly longer reads (PacBio and Oxford Nanopore, up to 10s of kilobases long), and accordingly, aligners to map such long reads are also of recent interest in the bioinformatics community.
>
> **[Q3] “Can the authors comment on how this paper is a good fit for ICLR, and the steps the author may take to make this paper more accessible to the ML audience?”**
>
> Representation learning holds a pivotal role within the framework of ICLR. The treatment of DNA sequences demands a computational approach grounded in deep learning, as evidenced by numerous recent works in the field [1,2,3]. Notably, the guidelines for ICLR explicitly outline a track dedicated to "Applications in biology, and related fields."
>
> In our endeavor to make our paper accessible to the broader ICLR community, which includes individuals with expertise in machine learning, we kindly direct the reviewer's attention to the first part of the response. We believe our work aligns with the conference's objectives, particularly within the specified track.
>
>
> [1] Nguyen, Eric, et al. "Hyenadna: Long-range genomic sequence modeling at single nucleotide resolution." arXiv preprint arXiv:2306.15794 (2023).
>
> [2] Dalla-Torre, Hugo, et al. "The nucleotide transformer: Building and evaluating robust foundation models for human genomics." bioRxiv (2023): 2023-01.
>
> [3] Ji, Yanrong, et al. "DNABERT: pre-trained Bidirectional Encoder Representations from Transformers model for DNA-language in genome." Bioinformatics 37.15 (2021): 2112-2120.

---

> > ### Comment · Reviewer_ModJ · 2023-11-23
> > **Thanks for the response**
> >
> > Dear authors,
> >
> > Thanks for the response. I do like the emphasis on semantic embedding in the general response above. I believe emphasizing and evaluating that in the paper will make it stronger. For example, the papers you cited, DNABERT, and nucleotide transformer perform well for standard benchmarks but the quality of embeddings is unclear. Specifically, for many downstream DNA tasks, specific and often rare patterns of DNA are important for classification, and if your paper can capture that better, it will be a significant result. My other comment on demonstrating the use in multiple downstream tasks is also important. ICLR indeed has a section for biology but it is important to make the paper more accessible than it currently is. For example, the papers HyenaDNA and nucleotide transformer have a fairly accessible Section 1.

---

> > > ### Author Response · Authors · 2023-11-23
> > >
> > > We thank the reviewer for reading the general comments and also encourage them to review the responses to other reviewers. There was not enough room in the general comments section to answer many of the shared set of concerns from all the reviewers. Some of the concerns specific to a reviewer are addressed in more detail in reply to similar concerns raised by another reviewer.
> > >
> > > *Specifically, the reviewer noted concerns about the evaluation method: simulated vs. real data. For this, kindly see our response to Reviewer 3, Q1.*
> > >
> > > We appreciate the reviewer’s agreement with us on the challenge that such LVL’s pose: one needs to capture sequence level distance metric much  more precisely that done in the case of LLMs. We believe that Figure 2 clearly demonstrates that existing Transformer-DNA models cannot directly perform sequence alignment since they do not implicitly model the precise SW distance metric.
> > >
> > > Re: the additional downstream tasks: We would like to note that Sequence Alignment, unlike the several downstream tasks that other Transformer-DNA models address, is a significantly larger task that on its own unlocks solving several subtasks -- such as the chromosome-specific and across-species alignment demonstrated in Table 2,3. We plan to apply our model to downstream classification tasks. As we note in the last paragraph of the paper (Concluding Remarks) and in the General Comments,  this model can be applied to alignment tasks concerning the Pan Genome and on a complementary task of Genome Assembly. This is because unlike traditional aligners, *the trained DNA-ESA model constructs a general mapping for sequences of nucleotides to a projection space* (irrespective of the reference from which it is derived). As demonstrated in Figure 1, global patterns in this space emerge naturally, and one can simply trace along these paths to reconstruct a reference genome.  We intend this paper to be a first step in such directions.
> > >
> > > Finally, we thank the reviewer for increasing their score and humbly request that they consider awarding a higher score given that we address most of the points that were brought up by them specifically, and even collectively by all the reviewers, in the revised version.

---

### Author Response · Authors · 2023-11-22

We are grateful  to the reviewers for their thorough evaluation, particularly for acknowledging the novelty of our method while pointing out how the method can be better justified and validated via further evaluations.  As emphasized in our revised version (see Background), our goal is to create a foundation model for Limited Vocabulary Languages (LVL), as in genomes, where the vocabulary comprises only four symbols (A,T,G,C). LVLs, as opposed to natural Large Language Models (LLMs), encode precise micro-level instructions into the exact order of symbols, and longer sequences correspond to functionality. For example, nature has preserved core genome sequences across species, and famously, Chimpanzees share 96% of the genetic codes with humans.

*Thus, an effective foundation model for generating **semantic embedding** of sequences drawn from LVLs faces a challenging task*: Euclidean space similarity metrics between embeddings of different length sequences **must be an accurate estimate** of the precise but computationally intensive similarity metrics defined on the original sequences themselves. The Smith-Waterman distance (SW) is an example of such a sequence-level metric that is used to compute the edit distance between a pair of genomic sequences. Moreover, given the consistency of genetic codes across species, our foundation model should also be able to semantically embed sequences for any species, even when trained only on the human genome.

As described in our manuscript, prior work on using the Transformer architecture to encode genomic sequences focused on obtaining representations that could be fine-tuned to solve classification tasks, where, *as in the case of LLMs, you need not preserve the pairwise inter-sequence distances*. There are, however, tasks in LVLs, such as Sequence Alignment, that require that a given subsequence be exactly matched to its location in a reference genome. For our DNA-ESA model to solve this task it has to achieve the following: given a query embedding of a short sequence, a list of top-$K$ of its nearest neighbors of pre-encoded fragment sequences **must contain** the best match as determined by sequence-level SW distance.

Our paper addresses this gap, and shows that such a foundation model is possible without ever explicitly using sequence level distances. Training is done using a contrastive loss that minimizes the cosine distance between the representations of a genomic sequence and a randomly-sampled subsequence. At the same time, the loss setup maximizes the distance between sequences that we select to have no overlap. We show that this signal alone is sufficient to transfer the SW distance metric to the representation space.

Conventional aligners such as -- Bowtie-2, Minimap-2, BWA-Mem2 -- use sequence-level information to pre-index the entire reference genome (exhaustively and efficiently), and given their development since the 1970s, it is not surprising that they achieve 99% recall success rate (see Table 4).

*In our work, we consider it a success that the first-generation DNA-ESA model is able to perform within 2-3% of these mature aligners*. The sequence alignment task serves as a first proof point of transferring a precise and computationally expensive distance metric over LVLs to a representation space. We expect this new capability will enable us to beat conventional methods on more difficult tasks such as genome assembly, where reads have to be assembled back into their correct locations, without access to any reference genome.
Below we list the changes to the new revision. Many of these points are shared across reviewers:

- DNA-ESA is validated on noisy ART reads: $Q_{PH} \in [10,20]$, $I,D = 1$% (see Table 4, Reviewer 3, Q1). We observe a performance within 1% of those reported in Table 1.
- Task transfer is extended to species (Thermus Aquaticus, AcidoBacteriota) unrelated to the human genome. We report high performance even at top-K=1 (see Reviewer 3, Q3)
- Long reads from PacBio CCS (real reads generated on Ashkenazim Trio, Son) are successfully aligned with DNA-ESA (see Table 4 and Reviewer 3, Q1).
- We compute additional metrics (Precision, Accuracy, F1) in addition to Recall in Table 2 to show that DNA-ESA is not only sensitive but also specific (refer to Reviewer 2, Q4, Table 2).
- Fig. 1 now contains both reads and fragments to demonstrate that in the 2D viz. space, the reads and fragments are located close to one other (see Response to Reviewer 2, Q1)
- The background is revised to highlight the motivation of the paper which is to offer *a proof point of the ability to transfer a precise and computationally expensive distance metric to a representation space.*
- Eq. 4 now includes the equality constraint (to account for homologs and repeats). The notation in Eq. 6 is clarified. (Kindly refer to Reviewer 2, Q2).

We encourage the reviewers to respond to comments as well as to kindly read through the revision. Thank you.

---

### Meta-Review · Program_Chairs · 2024-01-15

**Metareview:**

This paper proposes a transformer-based approach to address the DNA sequence alignment problem. The general idea is to train an encoder to capture the similarities between input short sequences and the long reference sequences. Then a kNN search is performed for a query sequence based on the embedding. Finally, an exact alignment algorithm is performed. The reviewers all consider this work has interesting ideas and experiments on simulated data show good accuracy. The concerns are: (1) the dataset is simulated, and real-world dataset is expected; (2) whether the similarity between embeddings are the good indicator of alignment; and (3) the contribution is not directly related to alignment but DNA encoder. The authors have made a good rebuttal, but have not addressed all the above concerns.

**Justification For Why Not Higher Score:**

The reviewers' ratings are leaning to reject, and their concerns are not fully addressed in the rebuttal.

**Justification For Why Not Lower Score:**

N/A

---

### Decision · Program_Chairs · 2024-01-16

Reject